# RegionDoc-R1: Reinforcing Semantic Layout-Aware Learning for Document Understanding

## Abstract

Recently, eliciting reasoning abilities in Multimodal Large Language Models (MLLMs) through rule-based Reinforcement Learning (RL) has proven promising. In this work, we introduce ***RegionDoc-R1***, a novel framework for document understanding that enhances MLLMs' reasoning with step-wise feedback. Directly applying RL training with the GRPO algorithm to document reasoning presents two primary challenges: (i) a lack of layout modeling for document understanding, and (ii) the scarcity of high-quality document-reasoning data. To address these issues, we first propose the Region-Aware Group Relative Policy Optimization (*RA-GRPO*), which encourages models to utilize region-level spatial information in documents for reasoning. Instead of previous OCR-based text positions, we incorporate high-quality semantic reasoning layout in documents, linking visual regions directly to question-answer semantics. Correspondingly, we construct a hybrid training corpus, named *SR-Doc*, containing Semantic Reasoning (SR) examples enriched with cross-page and region-level reasoning layout annotations. Meanwhile, we also introduce an Adaptive Chain-of-Thought (*Ada-CoT*) strategy, which dynamically adjusts the reasoning process according to different tasks, enabling more efficient and flexible step-wise document understanding. Experiments on several document reasoning benchmarks demonstrate that ***RegionDoc-R1*** achieves state-of-the-art performance across tasks such as form understanding, table-based QA, and layout-sensitive information extraction.

## 1 Introduction

Document understanding is a fundamental task at the intersection of Natural Language Processing (NLP) and computer vision, with applications in information retrieval Rajpurkar et al. (2016); Yang et al. (2018), question answering Hermann et al. (2015); Chen et al. (2017), enterprise knowledge management Li et al. (2020); Gu et al. (2021), information extraction from Visually Rich Documents Perot et al. (2024); Bhattacharyya et al. (2025), and legal or financial analysis Chalkidis et al. (2020); Kornilova & Eidelman (2019). Early approaches typically relied on Optical Character Recognition (OCR) to extract plain text from documents, which was then processed using standard NLP pipelines. However, OCR-only methods struggle to capture the rich multimodal nature of real-world documents, which often contain images, tables, and other structured visual elements. To overcome this limitation, researchers proposed treating entire documents as images and applying computer vision techniques, such as convolutional neural networks and vision transformers, to capture holistic visual cues Appalaraju et al. (2021); Garncarek et al. (2021); Xu et al. (2020a). These methods can effectively leverage visual signals such as font styles, graphical elements, or color usage. Nevertheless, they often fail to handle densely packed text regions or complex structured components like tables and figures, which demand fine-grained reasoning over both visual and textual information.

With the rise of Multimodal Large Language Models (MLLMs) Liu et al. (2023); Bai et al. (2023), recent work has attempted to jointly process textual and visual modalities for document understanding Ye et al. (2023c); Zhang et al. (2023). For instance, AlignVLM Masry et al. (2025) focuses on improving vision-language alignment to achieve stronger performance across diverse tasks, yet it overlooks the *layout structure*, which often leads to reasoning failures for document-level tasks, particularly when information must be read in a structured order rather than a linear sequence. Document layout plays a crucial role in conveying semantic relationships between elements such as headers, footnotes, columns, or table cells Xu et al. (2020b); Powalski et al. (2021). Beyond modeling

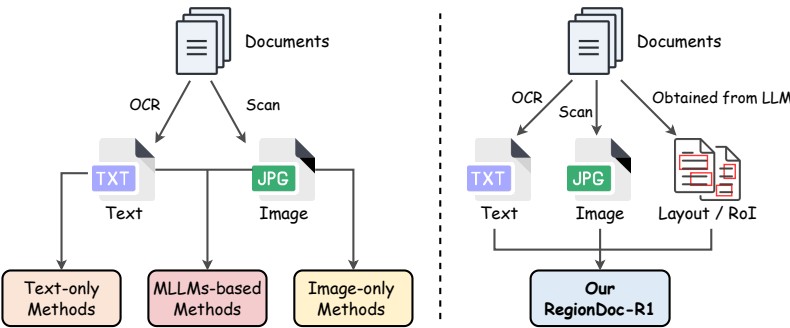

Figure 1: **Comparison of different paradigms in document understanding. Left**: Traditional pipelines obtain plain text via OCR or images via scanning, which are then processed by text-only, image-only, or MLLMs-based methods. **Right**: Our proposed ***RegionDoc-R1*** further incorporates semantic layouts and fine-grained Regions of Interest (RoI) obtained from LLMs, enabling region-level layout-aware reasoning across pages.

such relationships, layout information also enables *region-specific reasoning*, where only a localized portion of the document is relevant to a given question or task. This mirrors human reading strategies, in which readers typically locate the pertinent region before focusing on deeper understanding Lee et al. (2022); Garncarek et al. (2021).

In parallel, Reinforcement Learning (RL) has recently emerged as an effective paradigm for enhancing reasoning in LLMs. For example, DeepSeek-R1 Guo et al. (2025) demonstrates that large-scale RL can significantly boost Chain-of-Thought (CoT) reasoning and verification abilities. Beyond text-based reasoning, RL has also been applied in multimodal scenarios such as video understanding, where Video-R1 Feng et al. (2025) leverages preference optimization to improve temporal reasoning and action comprehension. These advances suggest that RL offers a powerful mechanism to align MLLMs toward more accurate and human-like reasoning. Recently, in the domain of document understanding, DocThinker Yu et al. (2025) employs rule-based strategies to improve explainability, while DocR1 Xiong et al. (2025) applies GRPO to enable multi-page reasoning. However, both approaches remain limited: DocThinker largely assumes evidence within a single page, and DocR1 operates at a coarse page level without precise localization of fine-grained Regions of Interest (RoI).

To address these limitations, we introduce ***RegionDoc-R1***, a region-aware reinforcement learning framework for semantic document reasoning. Unlike previous efforts, ***RegionDoc-R1*** introduces *Region-Aware Group Relative Policy Optimization (RA-GRPO)*, which leverages semantic layouts to precisely localize fine-grained ROIs, even when they span multiple pages and directly linked to question-answer semantics. This region-level perspective enables more accurate reasoning and better reflects human strategies of first identifying relevant areas and then performing focused understanding. To support this, we curate a new hybrid corpus, *SR-Doc*, which incorporates high-quality Semantic Reasoning (SR) layouts obtained from LLMs, and provides diverse instances with explicit multi-page and ROI-oriented annotations. In addition, we design an *Adaptive Chain-of-Thought (Ada-CoT)* mechanism that tailors the reasoning trajectory to different questions, thereby improving both precision and contextual awareness during reasoning. Together, these innovations equip ***RegionDoc-R1*** with the ability to integrate layout, semantics, and adaptive reasoning, thus advancing the state-of-the-art in various document understanding tasks.

Our main contributions are summarized as follows:

- We introduce ***RegionDoc-R1*** as a new reinforcement learning framework for *region-level, layout-aware* document reasoning within MLLMs, paving the way for more adaptable and robust document understanding.
- We propose a Region-Aware Group Relative Policy Optimization (*RA-GRPO*) method that exploits semantic layouts for reasoning, supported by a new hybrid dataset *SR-Doc* with cross-page RoI-structured annotations, and further enhanced by an Adaptive Chain-of-Thought (*Ada-CoT*) strategy for dynamic question-specific reasoning.
- We conduct extensive experiments on multiple benchmarks, showing that ***RegionDoc-R1*** outperforms existing methods and demonstrates the benefits of adaptive ROI-based RL for multi-page document understanding

## 2 RELATED WORK

### 2.1 MULTIMODAL DOCUMENT UNDERSTANDING

Large Language Models (LLMs) and Multimodal LLMs (MLLMs) have shown strong capabilities in understanding visually rich documents by leveraging both textual and layout information. Traditional methods such as LayoutLM series Xu et al. (2020b;a); Huang et al. (2022) incorporate layout and image features to improve performance in tasks such as form understanding and receipt parsing. Recently, instruction tuning models such as LLaVA-Doc Liu et al. (2023) further adapt LLMs to document understanding using multimodal data. mPLUG-DocOwl Ye et al. (2023a) incorporates multi-task instruction tuning over diverse document understanding datasets, including captioning, VQA, and information extraction. Unified models like UniDoc Gu et al. (2021) and UDOP Tang et al. (2023) generalize across various document tasks via a sequence-to-sequence formulation, enabling zero-shot and few-shot reasoning. More recently, LayoutLLM Luo et al. (2024) introduces a layout-aware tuning strategy that integrates document pre-trained models with fine-tuned LLMs. However, while prior works primarily focused on supervised pre-training or fine-tuning, the development of MLLMs with strong document reasoning capabilities through RL remains largely unexplored.

### 2.2 LARGE LANGUAGE MODEL REASONING

Reasoning with Large Language Models (LLMs) has attracted significant research interest due to their emerging capabilities in multi-step problem-solving. A key line of work focuses on *Chain-of-Thought (CoT)* prompting, which encourages models to explicitly decompose complex tasks into intermediate steps Wei et al. (2022); Kojima et al. (2022). This strategy has been shown to significantly improve performance in arithmetic, commonsense, and symbolic reasoning benchmarks. To scale up CoT data, automatic generation methods have been proposed Wang et al. (2023); Zhou et al. (2022), often relying on strong teacher models, such as GPT-4 Achiam et al. (2023). For example, LLaVA-CoT Xu et al. (2024) employs structured CoT templates (*e.g.*, summary, caption, reasoning, conclusion) for SFT, while Mulberry Yao et al. (2024) proposes CoMCTS, a collaborative Monte Carlo Tree Search method for discovering diverse reasoning paths. More recent efforts have investigated self-improvement via Reinforcement Learning (RL). DeepSeek-R1 Guo et al. (2025) shows that even coarse, outcome-only rewards can guide models to generate effective CoT reasoning via rule-based RL. Follow-up works such as Open Reasoner Zero Hu et al. (2025) and Kimi-1.5 Team et al. (2025) further validate this paradigm in both the text and vision domains. Extending RL to multimodal document understanding is a natural next step, especially when reasoning must incorporate both semantic and structural signals. However, applying such techniques to document understanding still faces challenges, such as effectively leveraging layout information.

### 2.3 LAYOUT MODELING IN DOCUMENT UNDERSTANDING

Modeling the spatial layout of documents is crucial for understanding structured information in visually rich documents. Early works such as LayoutLM Xu et al. (2020b), LayoutLMv2 Xu et al. (2020a), and LayoutLMv3 Huang et al. (2022) jointly encode textual content and 2D positions, while extensions like StrucTexT Li et al. (2021) and XYLayoutLM Gu et al. (2022) capture fine-grained spatial and hierarchical structures. OCR-free methods such as Donut Kim et al. (2022) and TILT Powalski et al. (2021) instead learn layout implicitly from raw pixels, though they struggle with explicit reasoning in multi-page or form-like documents. More recently, tuning-based multimodal models (*e.g.*, LLaVAR Zhang et al. (2023), LayoutLLM Luo et al. (2024)) incorporate layout cues indirectly, *e.g.*, through OCR-augmented captions or coarse layout embeddings. DocThinker Yu et al. (2025) introduces rule-based reinforcement learning for explainable reasoning but mainly assumes evidence within a single page, while DocR1 Xiong et al. (2025) applies GRPO to multi-page settings yet operates at a coarse page level without fine-grained region grounding. In contrast, our ***RegionDoc-R1*** introduces a layout-aware reinforcement learning framework that explicitly grounds reasoning in semantically relevant *region-level* RoIs across pages, enabling more accurate and robust document understanding. Unlike prior works where layout mainly comes from the spatial organization and reading order of content, our approach goes further by integrating task-specific semantic layouts that precisely identify the question-relevant RoIs, thus bridging spatial structure with semantic reasoning.

Figure 2: **Overview of the proposed *RegionDoc-R1* framework.** Given a document image $D$ and a question $q$, we provide ground-truth regions of interest (RoIs) $\mathcal{G}$ for SFT initialization to obtain the reference model $\pi_{\text{ref}}$. The policy model $\pi_\theta$ then predicts reasoning trajectories including intermediate thoughts, supporting RoIs, and final answers. Each predicted output is evaluated by our multi-objective reward functions (Sec. 3.2), including *Answer Accuracy Reward*, *Region Alignment Reward*, and *Adaptive CoT Reward*. The obtained rewards are aggregated and normalized into group-relative advantages, which are used to optimize the policy model via the $\mathcal{L}_{\text{RA-GRPO}}$ objective, while keeping it aligned with the reference model $\pi_{\text{ref}}$ through KL divergence regularization.

## 3 METHODOLOGY

Our goal is to develop a reinforcement learning framework that enhances region-aware reasoning for multimodal document understanding. While supervised fine-tuning (SFT) can endow models with general reasoning ability, it often fails to generalize robustly across diverse document layouts and tasks. To overcome these challenges, we propose ***RegionDoc-R1***, which augments SFT initialization with reinforcement learning guided by multiple reward signals. We explicitly link reasoning steps with semantically relevant Regions of Interest (RoIs) across pages. Beyond grounding, we further introduce an Adaptive Chain-of-Thought (Ada-CoT) mechanism to dynamically adjust reasoning traces according to different tasks, enabling both efficient and context-sensitive inference.

As shown in Figure 2, our method is based on GRPO (refer to Appendix A). Given a document and a question, the policy model predicts reasoning trajectories consisting of intermediate thoughts, localized RoIs, and final answers. Each candidate output is then evaluated by a set of complementary reward functions: (i) an *Answer Accuracy Reward* that verifies the correctness of final predictions, (ii) a *Region Alignment Reward* that enforces precise spatial grounding against annotated RoIs, and (iii) an *Adaptive CoT Reward* that regulates reasoning length and routing. The obtained rewards are normalized into group-relative advantages, which drive policy optimization through the objective, while a reference model obtained from supervised fine-tuning serves as a regularization anchor.

In the remainder of this section, we describe the construction of the SR-Doc dataset, the detailed design of our reward functions, and the overall optimization procedure of RA-GRPO.

### 3.1 SR-DOC DATASET

Existing datasets usually lack explicit region-level layouts and cross-page annotations, limiting their suitability for reinforcement learning in document understanding. To address this gap, we construct *SR-Doc*, a hybrid corpus designed for region-aware reinforcement learning. To address this gap, we construct *SR-Doc* by aggregating a broad range of existing document understanding datasets. For single-page documents, we use ten commonly used benchmarks, including DocVQA Mathew et al. (2021), InfoGraphicVQA Mathew et al. (2022), ChartQA Masry et al. (2022), DeepForm Svetlichnaya (2020), FigureQA Kahou et al. (2017), OCRVQA Mishra et al. (2019), TabFact Chen et al. (2019), TextCaps Sidorov et al. (2020), TextVQA Singh et al. (2019), and VisualMRC Tanaka et al. (2021). For multi-page documents, we curate three datasets: ArxivQA Li et al. (2024), DUDE Van Landeghem et al. (2023), and MP-DocVQA Tito et al. (2023).

To obtain high-quality region-level supervision, we adopt a two-step annotation strategy based on Qwen2-VL-72B-Instruct Wang et al. (2024). Given a document image and its QA pair from the

curated sources, we first prompt the model to identify the RoIs that are most relevant for answering the question. This ensures that the annotated regions are question-guided rather than generic visual spans. In the second step, we validate the annotation by cropping the predicted RoIs from the original document and re-feeding them to the model along with the question. If the model successfully produces the correct answer using only the selected regions, we retain this QA–RoI sample. This filtering step guarantees that each retained instance contains RoIs that are semantically meaningful and sufficient to answer the corresponding question, thereby ensuring the reliability of the annotation.

Overall, *SR-Doc* contains approximately **130k documents** with around **450k QA pairs**, aggregated from both single-page and multi-page sources. On average, each document spans **3.5 pages** and each QA pair is annotated with **4.1 RoIs**, reflecting the localized regions required to answer different questions. We divide the dataset into 120k training set and 10k dev set to validate the training stability.

In conclusion, *SR-Doc* introduces two main characteristics: **(1) Multi-page reasoning.** Many examples require integrating evidence across multiple pages, encouraging models to learn long-context reasoning. **(2) Region-level RoIs.** Instead of coarse page-level grounding, each question is paired with fine-grained RoIs directly linked to answer semantics, enabling precise layout-aware supervision.

## 3.2 REGION-AWARE GROUP RELATIVE POLICY OPTIMIZATION (RA-GRPO)

Standard GRPO Shao et al. (2024b) is a critic-free RL framework based on group-relative advantages, yet it overlooks spatial layout cues and assumes fixed-length reasoning traces. We therefore propose Region-Aware GRPO (RA-GRPO), which incorporates region-level layout supervision from *SR-Doc* and an Adaptive Chain-of-Thought (Ada-CoT) mechanism, enabling policies to answer accurately, ground their predictions in semantically relevant RoIs, and dynamically adjust reasoning strategies.

Given a question-document pair $(q, D)$, the policy model $\pi_\theta$ samples $K$ responses $\{(o^{(i)}, \hat{\mathcal{B}}^{(i)})\}_{i=1}^K$. Each candidate is scored by a multi-objective reward $R^{(i)}$, and the group-normalized advantage $A_i$ is:

$$A_i = \frac{R^{(i)} - \mu}{\sigma}, \qquad \mu = \tfrac{1}{K}\sum_{i=1}^K R^{(i)}, \quad \sigma = \sqrt{\tfrac{1}{K}\sum_{i=1}^K (R^{(i)} - \mu)^2 + \epsilon}. \tag{1}$$

**Answer Accuracy Reward.** First, we check whether the predicted final answer $o_{\text{final}}$ matches the ground-truth $y$. For *textual answers*, we directly check whether the predicted final answer $o_{\text{final}}$ matches the ground-truth $y$ after normalization:

$$R_{\text{ans}} = \begin{cases} 1, & \text{if } \text{norm}(o_{\text{final}}) = \text{norm}(y), \\ 0, & \text{otherwise.} \end{cases} \tag{2}$$

where $\text{norm}(\cdot)$ normalizes case and spacing. For *numeric answers*, we allow approximate matching by measuring the relative error $e$ with tolerance $\delta$:

$$e = \frac{|o_{\text{final}} - y|}{\max(|y|, 1)}, \qquad R_{\text{ans}} = \exp\!\left(-\tfrac{e}{\delta}\right). \tag{3}$$

**Region Alignment Reward.** To enforce spatial grounding, predicted RoIs $\hat{\mathcal{B}}$ are then compared with ground-truth RoIs $\mathcal{G}$. We compute IoU-based precision, recall, and $F_1$:

$$\text{Prec} = \frac{|\mathcal{M}|}{|\hat{\mathcal{B}}|}, \qquad \text{Rec} = \frac{|\mathcal{M}|}{|\mathcal{G}|}, \qquad F_1 = \frac{2 \cdot \text{Prec} \cdot \text{Rec}}{\text{Prec} + \text{Rec} + \varepsilon}, \tag{4}$$

where $\mathcal{M}$ is the set of matched box pairs under threshold $\tau$. We further measure region coverage:

$$\text{Cov} = \frac{\sum_p \text{Area}(\cup_{m:p_m=p}\hat{b}_m \cap \cup_{j:g_j.p=p}g_j)}{\sum_p \text{Area}(\cup_{j:g_j.p=p}g_j) + \varepsilon}, \tag{5}$$

where $\hat{b}_m$ and $g_j$ denote predicted and ground-truth RoIs on page $p$, respectively. This metric measures the fraction of ground-truth area covered by predictions, complementing $F_1$ by emphasizing

spatial completeness. The region reward $R_{\text{roi}}$ is:

$$R_{\text{roi}} = \alpha_1 F_1 + \alpha_2 \text{Cov} - \alpha_3 \cdot \frac{\max(|\hat{\mathcal{B}}| - |\mathcal{G}|, 0)}{\max(|\hat{\mathcal{B}}|, 1)}, \tag{6}$$

where $\alpha_1$ and $\alpha_2$ balance the contributions of $F_1$ accuracy and coverage, while $\alpha_3$ controls the penalty on *over-prediction*, *i.e.*, producing more RoIs than necessary.

**Adaptive CoT Reward.** We further introduce Ada-CoT to introduce dynamic control over whether and how reasoning traces are generated. The model first samples a routing variable $z \in \{\text{direct}, \text{cot}\}$:

$$p_\theta(z \mid q, D) = \text{softmax}(W^\top h(q, D)), \tag{7}$$

where $h(q, D)$ denotes the joint representation of the input question $q$ and document $D$, obtained from the hidden states of the underlying MLLM encoder. If $z = \text{direct}$, the model outputs an answer directly; if $z = \text{cot}$, it produces a reasoning trace $\tau$ consisting of $|\tau|$ steps.

Thus, we design two reward components:

$$R_{\text{gate}} = \begin{cases} 1, & \text{if } z = \text{direct} \text{ and prediction is correct,} \\ \kappa, & \text{if } z = \text{cot,} \\ 0, & \text{otherwise.} \end{cases} \tag{8}$$

Here $R_{\text{gate}}$ encourages the model to answer directly when possible, while still rewarding CoT routing with a base score $\kappa$.

$$R_{\text{len}} = \exp\left(-\frac{\max(|\tau| - B_0, 0)}{\delta_{\text{len}}}\right), \tag{9}$$

Where $R_{\text{len}}$ softly penalizes traces longer than $B_0$, while allowing shorter ones to retain full reward.

The overall Ada-CoT reward is defined as:

$$R_{\text{adacot}} = \rho_1 R_{\text{gate}} + \rho_2 R_{\text{len}}. \tag{10}$$

**Final Reward and Loss.** The overall reward is

$$R = \lambda_{\text{ans}} R_{\text{ans}} + \lambda_{\text{roi}} R_{\text{roi}} + \lambda_{\text{adacot}} R_{\text{adacot}}. \tag{11}$$

The RA-GRPO loss is then

$$\mathcal{L}_{\text{RA-GRPO}}(\theta) = -\frac{1}{K} \sum_{i=1}^{K} \frac{1}{|o^{(i)}|} \sum_{t=1}^{|o^{(i)}|} \left[ \frac{\pi_\theta(o_t^{(i)} \mid q, D, o_{<t}^{(i)})}{\varphi[\pi_\theta(o_t^{(i)} \mid q, D, o_{<t}^{(i)})]} \cdot A_i - \beta \, \mathbb{D}_{\text{KL}}(\pi_\theta \parallel \pi_{\text{ref}}) \right], \tag{12}$$

where $A_i$ is computed as in Eq. 1, and $\pi_{\text{ref}}$ denotes the reference policy (the SFT-initialized checkpoint), which serves as an anchor to regularize training via KL divergence. All hyperparameter settings and a detailed pseudo-code of our RA-GRPO are provided in Appendix B.

## 4 EXPERIMENTS

**Training.** We adopt a two-stage training strategy leveraging both existing multimodal reasoning resources and our newly constructed dataset. In the supervised fine-tuning (SFT) stage, we use the Visual CoT dataset Shao et al. (2024a), which contains 438k question-answer pairs with annotated bounding boxes highlighting key regions required to answer the questions. These annotations span five domains, including text/document understanding, fine-grained recognition, charts, general visual question answering, and relational reasoning. The bounding box supervision equips the model with generic multimodal reasoning skills and the ability to ground predictions in relevant visual regions.

In the reinforcement learning (RL) stage, we rely on our proposed *SR-Doc* dataset (Sec. 3.1), which provides multiple region-level RoI annotations over multi-page documents. Unlike Visual CoT, SR-Doc focuses specifically on document understanding tasks, offering direct spatial supervision

| Model | Param. | DocVQA | InfoVQA | TabFact | TextCaps | TextVQA | SROIE | VisualMRC |
|---|---|---|---|---|---|---|---|---|
| LLaVA-1.5-7B | 7B | 61.2 | 34.5 | 60.7 | 56.3 | 54.8 | 58.1 | 210.4 |
| VisCoT-7B | 7B | 69.7 | 39.8 | 65.9 | 58.2 | 61.7 | 62.4 | 221.6 |
| UReader | 7B | 66.8 | 43.1 | 68.9 | 59.8 | 58.7 | 61.2 | 223.5 |
| TextMonkey | 9B | 72.4 | 30.7 | 65.4 | 61.0 | 64.2 | 63.3 | 229.1 |
| DocOwl-1.5-Chat | 8B | 83.1 | 52.6 | **80.8** | 62.3 | 70.1 | 65.7 | 242.0 |
| DocOwl-2 | 8B | 81.5 | 48.2 | 77.4 | 60.5 | 67.5 | 66.9 | 219.3 |
| Qwen2.5-VL | 7B | 94.2 | 80.9 | 77.5 | 64.1 | 85.1 | 70.4 | **275.8** |
| DocThinker* | 7B | 80.2 | - | - | **75.7** | 83.6 | 81.4 | - |
| DocR1* | 7B | 95.1 | 82.6 | 79.6 | - | 81.0 | - | 251.6 |
| ***RegionDoc-R1*** | 7B | **95.3** | **83.2** | 79.9 | 75.1 | **85.8** | **81.7** | 252.4 |

Table 1: Comparison on seven single-page document reasoning benchmarks. The best results are highlighted in **bold**. Mark * means results are taken directly from the original paper. For DocThinker, we report the setting with $1536^2$ input resolution and 8K training data (highest reported in paper).

that aligns with our layout-aware reward design. This enables fine-grained policy optimization and improves the robustness of reasoning across diverse tasks.

**Evaluation.** We evaluate our model on a broad suite of document understanding datasets, grouped into *single-page* and *multi-page* settings. For single-page document reasoning, we include seven widely-used datasets: DocVQA (Task 1) Mathew et al. (2021), InfoGraphicVQA Mathew et al. (2022), TabFact Chen et al. (2019), TextCaps Sidorov et al. (2020), TextVQA Singh et al. (2019), SROIE Huang et al. (2019), and VisualMRC Tanaka et al. (2021). These datasets span diverse domains such as forms, charts, tables, and visually-rich natural images, providing a comprehensive evaluation for region-level document reasoning. To test cross-page reasoning ability, we adopt SlideVQA Tanaka et al. (2023), ArxivQA Xiong et al. (2025), DUDE Van Landeghem et al. (2023), and MP-DocVQA Mathew et al. (2021). These corpora feature long documents (*e.g.*, scientific articles, enterprise reports) where evidence is distributed across pages, stressing cross-page grounding and layout-aware reasoning.

**Implementation Details.** Our ***RegionDoc-R1*** is built on top of the Qwen2.5-VL 7B backbone. Training is conducted on 8 NVIDIA H100 GPUs with mixed-precision (FP16) optimization. In the SFT stage, we train on the Visual CoT dataset for 3 epochs with a batch size of 128 (16 per GPU). Images are resized to a maximum resolution of 896×896 while preserving aspect ratio, and textual inputs are truncated to 512 tokens. We use AdamW optimizer with a learning rate of 2e-5, a cosine decay schedule, and weight decay of 0.01. For the RL stage, we fine-tune on our *SR-Doc* for 1.5 epochs with the batch size of 64, due to the heavier reward computation. The maximum input resolution is set to 1024×1024 to better capture fine-grained RoIs. We adopt the same AdamW optimizer with a peak learning rate of 1e-5. Gradient clipping of 1.0 is applied to stabilize policy updates. All experiments are implemented in PyTorch and Hugging Face Transformers. Training takes approximately 6 days for the SFT stage and 4 days for the RL stage.

## 4.1 MAIN RESULTS

**Quantitative Results.** We evaluate ***RegionDoc-R1*** against recent state-of-the-art multimodal LLMs on both single-page and multi-page document reasoning benchmarks.

As shown in Table 1, ***RegionDoc-R1*** achieves strong results across all seven single-page benchmarks. Compared with LLaVA-1.5-7B Liu et al. (2024) and VisCoT-7B Shao et al. (2024a), our approach yields clear gains on layout-intensive tasks such as DocVQA and InfoVQA. Although DocOwl series Ye et al. (2023b); Hu et al. (2024) and Qwen2.5-VL Wang et al. (2024) are competitive, they mainly rely on image-text alignment rather than explicit region grounding. Our layout-aware RL provides an advantage on datasets like SROIE and InfoVQA, where precise RoI localization is critical. While DocThinker and DocR1 leverage RL, the former uses rule-based strategies and the latter stays at page-level granularity. By contrast, our fine-grained RoI grounding explains the improvements on TextVQA and SROIE. On TabFact, where reasoning is largely textual, DocOwl-1.5 remains slightly ahead, showing the complementarity of factual reasoning and spatial grounding.

Table 2 shows that ***RegionDoc-R1*** achieves the best results across all four multi-page benchmarks. Compared with Qwen2.5-VL, our method yields consistent gains, especially on ArxivQA, where

| Model | Param. | MP-DocVQA | DUDE | SlideVQA | ArxivQA |
|---|---|---|---|---|---|
| LLaVA-1.5-7B | 7B | $39.12_{\pm 0.25}$ | $25.20_{\pm 0.18}$ | $28.96_{\pm 0.22}$ | $5.40_{\pm 0.10}$ |
| VisCoT-7B | 7B | $49.61_{\pm 0.28}$ | $31.20_{\pm 0.25}$ | $45.52_{\pm 0.26}$ | $10.50_{\pm 0.14}$ |
| DocOwl-2 | 8B | $68.15_{\pm 0.32}$ | $32.02_{\pm 0.21}$ | $29.53_{\pm 0.19}$ | $7.19_{\pm 0.11}$ |
| Qwen2.5-VL | 7B | $87.44_{\pm 0.35}$ | $52.85_{\pm 0.27}$ | $70.32_{\pm 0.33}$ | $29.60_{\pm 0.22}$ |
| DocR1* | 7B | 87.45 | 54.39 | 71.96 | - |
| ***RegionDoc-R1*** | 7B | $\mathbf{88.32}_{\pm 0.34}$ | $\mathbf{55.71}_{\pm 0.29}$ | $\mathbf{73.45}_{\pm 0.35}$ | $\mathbf{42.57}_{\pm 0.25}$ |

Table 2: Performance comparison on four multi-page document reasoning benchmarks. *Results of DocR1 are directly taken from their paper. The best results are highlighted in **bold**.

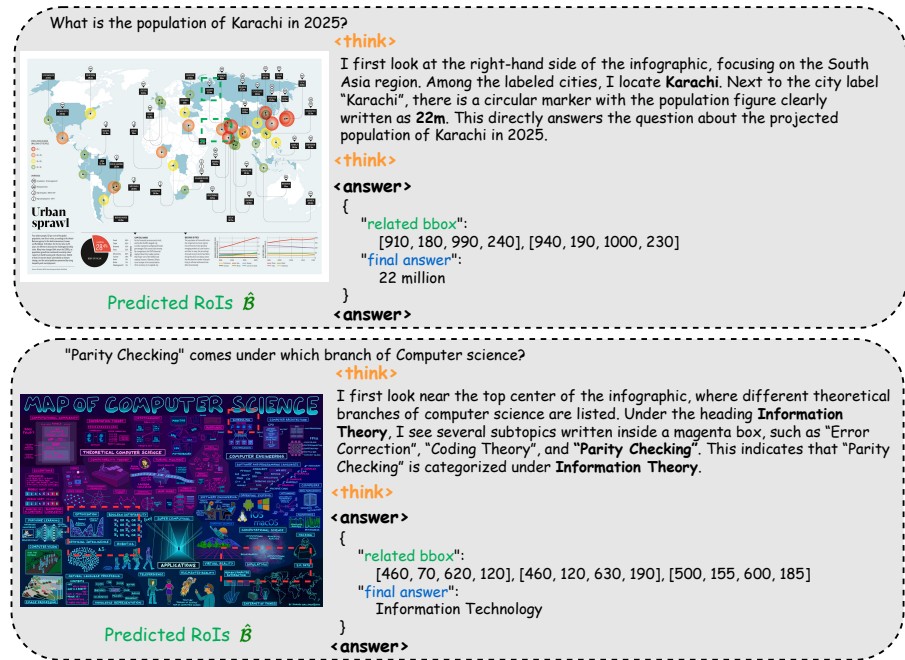

Figure 3: **Challenging cases of *RegionDoc-R1*.** The thinking process significantly improves the reasoning ability and explainability. Please zoom in to see the details.

region-level RoI grounding provides more precise supervision for long scientific articles. On MP-DocVQA and DUDE, the improvements demonstrate the effectiveness of layout-aware rewards in capturing cross-page dependencies. For SlideVQA, Ada-CoT helps balance direct answering and reasoning, avoiding unnecessarily long traces. Overall, these results validate that our region-aware RL framework enables stronger multi-page reasoning than prior page-level or rule-based approaches.

**Qualitative Results.** Figure 3 shows our method consistently identifies accurate RoIs and produces coherent reasoning traces across diverse document types. These visual examples highlight how region-level grounding and adaptive CoT jointly enable precise and interpretable document understanding.

### 4.2 ABLATION STUDY

**SFT *vs.* GRPO *vs.* RA-GRPO.** To better understand the contribution of our proposed RA-GRPO framework, we conduct ablation experiments comparing three settings: (1) **SFT only**, where the model is trained with supervised fine-tuning on VisualCoT without RL; (2) **GRPO**, where reinforcement learning is applied with standard group-relative rewards but without region-level or adaptive reasoning signals; (3) **RA-GRPO**, our full model with region-aware and adaptive rewards.

The results in Table 3 show three clear trends. First, reinforcement learning (GRPO) improves over SFT-only training, confirming the benefit of sampling-based optimization. Second, our RA-GRPO achieves the highest performance across all benchmarks, with consistent gains over standard GRPO. These improvements validate the importance of region-level layout rewards and adaptive CoT, which provide more informative supervision than simple correctness checks.

| Method | DocVQA | InfoVQA | MP-DocVQA | Avg. |
|--------|--------|---------|-----------|------|
| SFT only | 81.4 | 56.2 | 71.0 | 69.5 |
| GRPO | 84.7 | 60.1 | 74.2 | 73.0 |
| RA-GRPO (ours) | **88.3** | **63.5** | **77.6** | **76.5** |

Table 3: Ablation study comparing supervised fine-tuning (SFT), standard GRPO, and our RA-GRPO framework. Results are averaged across representative single- and multi-page benchmarks. For efficiency, we report results from shorter ablation runs, the training epochs and settings are kept consistent across variants.

**Ablation on Reward Components.** To further investigate the contribution of each reward, we conduct ablation experiments by selectively removing one component from our RA-GRPO framework: (1) w/o Answer Reward ($R_{ans}$), (2) w/o Region Reward ($R_{roi}$), (3) w/o Ada-CoT Reward ($R_{adacot}$). Results are reported on representative single-page (DocVQA) and multi-page (MP-DocVQA) benchmarks.

| Setting | DocVQA | MP-DocVQA | Avg. |
|---------|--------|-----------|------|
| RA-GRPO (full) | **88.3** | **77.6** | **83.0** |
| w/o $R_{ans}$ | 80.5 | 71.2 | 75.9 |
| w/o $R_{roi}$ | 84.1 | 73.5 | 78.8 |
| w/o $R_{adacot}$ | 86.0 | 75.1 | 80.6 |

Table 4: Ablation study on different reward components. Removing any single reward leads to performance degradation, showing their complementary benefits.

From Table 4, we observe that: (i) Removing $R_{ans}$ causes the largest drop, highlighting the necessity of explicit correctness supervision. (ii) Without $R_{roi}$, performance on MP-DocVQA decreases significantly, showing the importance of spatial grounding for multi-page reasoning. (iii) Excluding $R_{adacot}$ also reduces accuracy, especially on DocVQA, indicating that dynamic reasoning control helps avoid unnecessary long traces and improves efficiency. Together, these results demonstrate that all three rewards contribute complementary signals to robust document reasoning.

**Ablation on *SR-Doc*.** We also study the effect of our proposed SR-Doc dataset in reinforcement learning. To this end, we replace SR-Doc with the VisualCoT dataset Shao et al. (2024a) in the RL stage, while keeping the SFT initialization the same.

| Training Data for RL | DocVQA | MP-DocVQA | Avg. |
|---------------------|--------|-----------|------|
| VisualCoT (default) | 86.1 | 79.4 | 82.8 |
| SR-Doc (ours) | **95.3** | **88.3** | **91.8** |

Table 5: Ablation study on the effect of SR-Doc dataset during RL training. Using SR-Doc leads to consistent improvements, especially in multi-page settings.

As shown in Table 5, replacing SR-Doc with VisualCoT leads to clear performance degradation, particularly on MP-DocVQA. This is expected since VisualCoT mainly provides single-page visual reasoning with limited layout variety, lacking fine-grained RoI annotations across pages. By contrast, SR-Doc introduces region-level supervision and multi-page coverage, which enables RA-GRPO to exploit spatial cues and align reasoning with document structure. These results confirm that SR-Doc is essential for achieving robust multi-page and layout-aware document understanding.

## 5 CONCLUSION

We presented *RegionDoc-R1*, a novel reinforcement learning framework for document understanding that integrates layout awareness and adaptive reasoning into MLLMs. Our approach introduces *RA-GRPO*, which grounds reasoning at the region level with explicit spatial supervision, supported by the *SR-Doc* dataset containing fine-grained multi-page annotations. In addition, the *Ada-CoT* strategy enables dynamic adjustment of reasoning depth, balancing direct answering and step-wise inference. Experiments on diverse single- and multi-page benchmarks show that *RegionDoc-R1* achieves state-of-the-art performance, and demonstrates the effectiveness of combining layout-aware RL with adaptive reasoning for robust and interpretable document understanding.

## ETHICS STATEMENT

Our research focuses on developing reinforcement learning methods for document understanding. All datasets used in this paper, including DocVQA, InfoVQA, TabFact, TextVQA, DUDE, MP-DocVQA, and others, are publicly available benchmarks released under academic or research licenses. Our constructed *SR-Doc* dataset is built entirely from these open resources, and no private or personally identifiable data is involved. We ensure that all data processing follows the licenses of the source datasets, and any redistributed annotations will be shared under the same terms to support reproducibility. Future dataset releases will be carefully documented, with clear licenses and guidelines for responsible use.

Our methodology is intended solely for advancing multimodal reasoning in scientific and practical document understanding tasks (*e.g.*, forms, contracts, tables). It does not target sensitive domains such as surveillance, biometric identification, or generation of deceptive content. Nevertheless, as with all machine learning models, potential risks include unintended biases inherited from the training datasets and over-reliance on spurious correlations. We mitigate these risks by evaluating across diverse benchmarks and explicitly grounding reasoning in transparent RoI annotations.

We believe our work raises no direct ethical concerns related to human subjects, security, or privacy.

## REPRODUCIBILITY STATEMENT

We have taken several steps to ensure the reproducibility of our work. All implementation details, including model architecture, training procedure, and optimization settings, are provided in the main paper (Sec. 3) and Appendix. A complete list of hyperparameter settings used in RA-GRPO training is included in the Appendix for transparency. Our proposed SR-Doc dataset will be released under the same academic terms as its source datasets, and all processing steps are described in Sec. 3.1. To further facilitate replication and follow-up research, we have created an anonymous repository at https://anonymous.4open.science/r/RegionDoc-R1-F7DD, including code, pre-processing scripts, and pretrained checkpoints.

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

# A PRELIMINARY

**Group Relative Policy Optimization (GRPO).** Group Relative Policy Optimization (GRPO) Shao et al. (2024b) is a critic-free reinforcement learning algorithm designed for large-scale language models. Unlike classical methods such as Proximal Policy Optimization (PPO) Schulman et al. (2017), which rely on a value network to estimate returns, GRPO avoids the instability and computational burden of training a critic by comparing sampled responses within a group.

Formally, given a question $q$, the policy $\pi_\theta$ generates $K$ candidate responses:

$$\mathcal{O} = \{o^{(1)}, o^{(2)}, \ldots, o^{(K)}\}, \qquad o^{(i)} = (o_1^{(i)}, o_2^{(i)}, \ldots, o_{|o^{(i)}|}^{(i)}). \tag{13}$$

Each response $o^{(i)}$ receives a scalar reward $R^{(i)} \in \mathbb{R}$ from a task-specific rule-based evaluator:

$$R^{(i)} = R(q, o^{(i)}). \tag{14}$$

**Relative Advantage.** Instead of directly using $R^{(i)}$, GRPO computes a group-normalized *advantage* for stability:

$$\mu = \frac{1}{K} \sum_{i=1}^{K} R^{(i)}, \qquad \sigma = \sqrt{\frac{1}{K} \sum_{i=1}^{K} (R^{(i)} - \mu)^2 + \epsilon}, \tag{15}$$

$$A_i = \frac{R^{(i)} - \mu}{\sigma}, \qquad i = 1, \ldots, K, \tag{16}$$

where $\mu$ and $\sigma$ are the mean and standard deviation of the group, and $\epsilon$ is a stabilizer. This ensures that the learning signal depends on relative performance inside the group, not on absolute reward scales.

**Token-level Credit Assignment.** Each response $o^{(i)}$ is an autoregressive sequence, so its group-level advantage $A_i$ is broadcast to each decoding step:

$$A_{i,t} = A_i, \qquad t = 1, \ldots, |o^{(i)}|. \tag{17}$$

This means every token in $o^{(i)}$ shares the same sequence-level advantage.

**Policy Update.** The GRPO objective encourages high-probability assignment to better-than-average responses while regularizing against a reference policy $\pi_{\text{ref}}$ (the SFT checkpoint). The loss is:

$$\mathcal{L}_{\text{GRPO}}(\theta) = -\frac{1}{K} \sum_{i=1}^{K} \frac{1}{|o^{(i)}|} \sum_{t=1}^{|o^{(i)}|} \left( \underbrace{\frac{\pi_\theta(o_t^{(i)} \mid q, o_{<t}^{(i)})}{\varphi[\pi_\theta(o_t^{(i)} \mid q, o_{<t}^{(i)})]}}_{\text{policy ratio}} \cdot A_{i,t} - \beta\, \mathbb{D}_{\text{KL}}(\pi_\theta \,\|\, \pi_{\text{ref}}) \right). \tag{18}$$

Here, $\varphi[\cdot]$ denotes stop-gradient, $\beta$ controls KL regularization, and $\pi_{\text{ref}}$ stabilizes training by anchoring updates to the SFT-initialized model.

**Training Procedure.** In practice, GRPO is typically used in a two-stage pipeline. First, the model is initialized by supervised fine-tuning (SFT) on annotated corpora, providing a stable reference policy $\pi_{\text{ref}}$ and preventing collapse at the start of RL training. Then, reinforcement learning with GRPO is applied: for each input $(q, D)$, the model samples multiple candidate outputs $\{o^{(i)}\}_{i=1}^{K}$, computes group-normalized advantages as in Eq. (3), and updates the policy according to Eq. (6). The KL term anchors $\pi_\theta$ close to $\pi_{\text{ref}}$, ensuring that exploration does not drift too far from the supervised initialization.

This training paradigm allows models to iteratively refine their reasoning ability: SFT provides general-purpose generation skills, while GRPO emphasizes selecting the most promising responses within each sampled group. Such a framework has proven effective in scaling reasoning-centric large language models, and in our work, it provides the foundation for extending reinforcement learning to region-aware, layout-sensitive document reasoning.

**Summary.** GRPO thus replaces critic estimation with group-wise comparison, yielding three benefits: (1) critic-free stability, (2) robustness to reward scaling, and (3) scalability for LLM training. These properties make GRPO a natural foundation for our extension, RA-GRPO, where task-specific multi-objective rewards further incorporate spatial layout grounding and adaptive reasoning strategies.

# B    RA-GRPO TRAINING ALGORITHM

To explicitly ground reasoning in semantically relevant document regions, our RA-GRPO extends GRPO by incorporating multi-objective rewards: answer correctness, region alignment, and reasoning quality. The pseudo code in Algorithm 1 summarizes the entire training loop.

For reproducibility, we summarize the hyperparameter settings used in RA-GRPO training:

- **Sampling:** group size $K = 8$.
- **Answer Reward** $R_{\text{ans}}$**:** tolerance $\delta = 0.05$ for numeric answers; stability $\varepsilon = 10^{-8}$.
- **Region Reward** $R_{\text{roi}}$**:** IoU threshold $\tau = 0.5$; weights $(\alpha_1, \alpha_2, \alpha_3) = (0.5, 0.4, 0.1)$; stability $\varepsilon = 10^{-8}$.
- **Ada-CoT Reward** $R_{\text{adacot}}$**:** base CoT reward $\kappa = 0.5$; length threshold $B_0 = 8$; decay factor $\delta_{\text{len}} = 3.0$; weights $(\rho_1, \rho_2) = (0.7, 0.3)$.
- **Final Reward Aggregation:** weights $(\lambda_{\text{ans}}, \lambda_{\text{roi}}, \lambda_{\text{adacot}}) = (0.4, 0.4, 0.2)$.
- **Loss Regularization:** KL penalty $\beta = 0.1$; advantage stability $\epsilon = 10^{-8}$.

---

**Algorithm 1** Region-Aware Group Relative Policy Optimization (RA-GRPO)

---

**Input:** Document $D$, question $q$, ground-truth answer $y$, ground-truth RoIs $\mathcal{G}$
**Init:** Policy model $\pi_\theta$, reference policy $\pi_{\text{ref}}$, group size $K$
**for** *each training step* **do**
    // Step 1:  Sample candidate responses
    Sample $K$ responses $\{(o^{(i)}, \hat{\mathcal{B}}^{(i)}, z^{(i)}, \tau^{(i)})\}_{i=1}^{K}$ from $\pi_\theta$
    // Step 2:  Compute rewards for each candidate
    **for** $i = 1$ **to** $K$ **do**
        // Answer Accuracy Reward
        **if** $y$ *is textual* **then**
            $R_{\text{ans}}^{(i)} = \mathbb{1}[\text{norm}(o_{\text{final}}^{(i)}) = \text{norm}(y)]$
        **else**
            $e = \frac{|o_{\text{final}}^{(i)} - y|}{\max(|y|, 1)}$   $R_{\text{ans}}^{(i)} = \exp(-e/\delta)$
        // Region Alignment Reward
        Compute precision, recall, and $F_1$ by IoU-matching $\hat{\mathcal{B}}^{(i)}$ with $\mathcal{G}$  Compute coverage Cov
          across pages   $R_{\text{roi}}^{(i)} = \alpha_1 F_1 + \alpha_2 \text{Cov} - \alpha_3 \cdot \frac{\max(|\hat{\mathcal{B}}^{(i)}| - |\mathcal{G}|, 0)}{\max(|\hat{\mathcal{B}}^{(i)}|, 1)}$
        // Adaptive CoT Reward
        **if** $z^{(i)} = \text{direct}$ *and correct* **then**
            $R_{\text{gate}}^{(i)} = 1$
        **else**
            $R_{\text{gate}}^{(i)} = \kappa$
        $R_{\text{len}}^{(i)} = \exp(-\frac{\max(|\tau^{(i)}| - B_0, 0)}{\delta_{\text{len}}})$   $R_{\text{adacot}}^{(i)} = \rho_1 R_{\text{gate}}^{(i)} + \rho_2 R_{\text{len}}^{(i)}$
        // Final Reward
        $R^{(i)} = \lambda_{\text{ans}} R_{\text{ans}}^{(i)} + \lambda_{\text{roi}} R_{\text{roi}}^{(i)} + \lambda_{\text{adacot}} R_{\text{adacot}}^{(i)}$
    // Step 3:  Compute group-normalized advantages
    $\mu = \frac{1}{K} \sum_i R^{(i)}$, $\sigma = \sqrt{\frac{1}{K} \sum_i (R^{(i)} - \mu)^2 + \epsilon}$  $A_i = (R^{(i)} - \mu)/\sigma$ for each $i$
    // Step 4:  Policy update
    Update $\pi_\theta$ by minimizing:

$$\mathcal{L}_{\text{RA-GRPO}} = -\frac{1}{K} \sum_{i=1}^{K} \frac{1}{|o^{(i)}|} \sum_{t=1}^{|o^{(i)}|} \left( \frac{\pi_\theta(o_t^{(i)} \mid q, D, o_{<t}^{(i)})}{\varphi[\pi_\theta(o_t^{(i)} \mid q, D, o_{<t}^{(i)})]} \cdot A_i - \beta \, \mathbb{D}_{\text{KL}}(\pi_\theta \,\|\, \pi_{\text{ref}}) \right)$$

---

## THE USE OF LARGE LANGUAGE MODELS (LLMS)

This work makes use of large language models (LLMs) in two ways. First, our proposed framework builds upon an existing multimodal LLM backbone (Qwen2.5-VL-7B), which we further tune using supervised and reinforcement learning. Second, during the construction of the *SR-Doc* dataset, we employed Qwen2-VL-72B-Instruct to generate region-level annotations by identifying question-relevant RoIs within multi-page documents. The outputs were subsequently filtered and validated to ensure quality. Beyond these roles, no LLMs were used for research ideation, writing, or analysis.

