# OpenReview forum: "RegionDoc-R1: Reinforcing Semantic Layout-Aware Learning for Document Understanding"
_ICLR.cc/2026/Conference — ICLR 2026 Conference Withdrawn Submission_

### Official Review · Reviewer_Ve2p · 2025-10-27

**Soundness:** 3
**Presentation:** 3
**Contribution:** 3
**Rating:** 4
**Confidence:** 4

**Summary:**

The paper presents RegionDoc-R1, a reinforcement learning framework for document understanding that incorporates region-aware group relative policy optimization (RA-GRPO). To address the lack of spatial supervision and reasoning data, the authors construct SR-Doc, a dataset with fine-grained region annotations and cross-page semantic reasoning labels. RegionDoc-R1 further introduces an Adaptive Chain-of-Thought (Ada-CoT) strategy for dynamic reasoning. Experiments on multiple benchmarks demonstrate strong performance and clear logic in both method and presentation.

**Strengths:**

The paper constructs a new dataset (SR-Doc) with detailed region-level annotations, proposes RegionDoc-R1 with a novel region-aware reward design, and evaluates the model thoroughly on both single-page and multi-page benchmarks. The experimental results and ablations are clear, showing the method’s effectiveness.

**Weaknesses:**

1. The performance on VisualMRC drops by more than 8% compared to Qwen2.5-VL, but the paper doesn’t discuss why this happens. It would help to include some analysis or error cases.
2. In the ablation study (Table 3 and Table 4), GRPO actually performs better than the “w/o $R_{\text{roi}}$” variant on both DocVQA and MP-DocVQA, which suggests that the adaptive reasoning reward ($R_{\text{adacot}}$) might make training a bit unstable when used alone.
3. All the qualitative examples are single-page, so the paper doesn’t really show how the method handles cross-page reasoning, which is supposed to be one of its main strengths.

**Questions:**

1. The model’s performance on VisualMRC drops by over 8% compared to Qwen2.5-VL. Any insight why this happens?
2. In Table 3 and 4, GRPO actually performs better than the “w/o $R_{\text{roi}}$” variant. Could you clarify whether the GRPO baseline uses only the answer correctness reward ($R_{\text{ans}}$)? If so, why the adaptive reasoning reward ($R_{\text{adacot}}$) hurts stability?
3. Since multi-page reasoning is one of the main focuses of the paper, could you show a qualitative case for that setting?

One suggestion: More dataset context (including training and testing datasets) would also help readers interpret results. Like the average document length, number of pages, or how many pages typically contain evidence.

I would be willing to increase the score if the questions are answered.

---

### Official Review · Reviewer_ZPf9 · 2025-10-31

**Soundness:** 3
**Presentation:** 3
**Contribution:** 2
**Rating:** 4
**Confidence:** 3

**Summary:**

This paper presents RegionDoc-R1, a reinforcement learning framework for document understanding that enhances MLLMs with region-aware reasoning and adaptive step-wise feedback. Experimental results on both single-page and multi-page document reasoning benchmarks demonstrate that RegionDoc-R1 outperforms existing methods, achieving improved reasoning grounded in document layout.

**Strengths:**

- This paper is clearly written and well-structured.

- The work proposes a novel methodology by introducing the Answer Accuracy Reward, Region Alignment Reward, and Adaptive CoT Reward into GRPO, effectively enhancing reasoning in document understanding.

- Extensive experiments across multiple benchmarks, including single-page and multi-page document reasoning, demonstrate state-of-the-art performance. Comprehensive ablation studies further validate the effectiveness of RA-GRPO, its reward components, and the SR-Doc dataset

**Weaknesses:**

- Dependence on handcrafted reward design: The effectiveness of RA-GRPO heavily relies on manually tuned reward weights and thresholds (e.g., IoU , λ ...) without providing sensitivity analysis or justification for these hyperparameters. This raises concerns about robustness and scalability, as careful hyperparameter adjustment may be needed for different document domains or reasoning tasks.

- Limited analysis of adaptive reasoning dynamics: While Adaptive CoT is introduced to balance direct answering and multi-step reasoning, the paper lacks quantitative or qualitative analysis of how Ada-CoT affects reasoning behavior, such as routing patterns, reasoning length distribution, or interpretability.

**Questions:**

- Could the authors clarify how hyperparameters are selected? A sensitivity analysis showing how variations in these hyperparameters affect model performance would be helpful.

- Could the authors explain the rationale behind the Adaptive CoT reward design? According to Equation 8, direct answers receive a reward of 1.0, while CoT receives 0.5, which may implicitly discourage the model from generating explicit reasoning chains. How does this design ensure that the model still performs multi-step reasoning when necessary?

- Since the SR-Doc annotations and validation are generated automatically using Qwen2-VL-72B-Instruct, how do the authors ensure the accuracy and human consistency of the reward signals? Have any small-scale human evaluations been conducted on the RoI annotations in SR-Doc?

---

### Official Review · Reviewer_Mgen · 2025-10-31

**Soundness:** 2
**Presentation:** 2
**Contribution:** 2
**Rating:** 4
**Confidence:** 4

**Summary:**

The paper describes a framework for doucment understanding based on reinforcement learning, where the main contribution comes from introducing region-aware information in the reward function to guide the model towards relevant regions to answer the question. It also introduces a chain-of-thought reward to guide the generation of reasoning steps. In order to be able to train the model with region information, the paper also introduces a new dataset derived from a set of existing datasets, automatically annotated with the relevant regions required to answer the question. The model is evaluated on a collection of standard benchmarks for text-based VQA, both single and multi-page.

**Strengths:**

- The paper introduces a new dataset for text-based VQA annotated with the set of relevant regions necessary to answer the question.
- The paper introduces a reward function to add region-aware guidance to the standard reinforcement learning with GRPO in order to guide the model towars relevant regions for the question.

**Weaknesses:**

- The initial claims of the paper are a bit misleading. The proposed method is not integrating layout modeling (usually layout modeling in document understanding refers to identifying the different sections/elements of the document), but rather using region aware guidance to answer the question. In that sense figure 1 is misleading since, as far as I understand, ROIs are not an input to the model, but an output of the model along with the answer to the question. The figure also suggests that OCR is an additional input to the model, but this is not clarified in the paper. Is OCR an input to the model. In that case, how OCR is obtained?
- Concerning the creation of the dataset, some of the claims is also misleading. In particular, at the end of section it is claimed that SR-Doc introduces multi-page reasoning. However, this is a characteristic that is not introduced by the proposed dataset, but it derives from the original datasets that already require multi-page reasoning.
- Concerning the process of annotation and validation of the dataset, it includes a validation to check that the question can be answered using the identified ROIs, but there is no validation to check that this is the minimal region needed to answer the question or that the model can answer without looking at the selected regions, just using some prior knowledge or bias.
- Concerning the design of the policy, the selected metric for textual answers is accuracy while in DocVQA usually ANLS is used to allow for some misspelling in the answer due to OCR errors. I think it would make sense to consider ANLS in this case. On the other hand, for numbers some error is allowed when computing the reward. However, as most datasets contain extractive answers that do not require any arithmetic operation, probably it would make more sense to use exact matching or ANLS again. For the region-aware reward, defining the third term that controls over-prediction as a function of the number of predicted regions can be inaccurate. I think it would be better to define it as a function of the global overlapping or coverage between prediction and GT. Finally, with respect to the CoT reward it is not clear the rationale behind prioritizing direct and short answers and why is the motivation of calling it adaptive Chain-of-Thought. I do not see the relation between the name and the definition.
- Results in table 1 show only some slight improvement with respect to DocR1 (in some cases, even some performance drop). Thus, the advantage of the proposed method over SoA seems limited. In particular, as far as I understand, the main difference between DocR1 and the proposed method are the region-aware and CoT rewards. These results question the contribution of the proposed new rewards.
- The results of the proposed method in tables 3 and 4 with the ablation study are lower than those in tables 1 and 2. As suggested by the caption in table 3 this could be due to shorter ablation runs. However, this could distort the results of the ablation study. For instance with full training the difference between DocR1 and the proposed method is lower than the difference in the ablation study between the configuration with basic GRPO (as it is supposed to be in DocR1) and the configuration with region and CoT rewards.

**Questions:**

N/A

---

### Note · Authors · 2025-11-12

I have read and agree with the venue's withdrawal policy on behalf of myself and my co-authors.